# OpenReview forum: "Beyond Generic Benchmarks: Evaluating the Structural Misalignment of LLMs in Public-Sector Decision Contexts"
_ICLR.cc/2026/Conference — Submitted to ICLR 2026_

### Official Review · Reviewer_6eiW · 2025-10-31

**Soundness:** 2
**Presentation:** 3
**Contribution:** 3
**Rating:** 4
**Confidence:** 4

**Summary:**

This work focuses on the research question that generic LLM benchmarks tend to overlook domain-specific and population-level harms in high-stakes public sector settings. The authors address this question specifically in child maltreatment by constructing an evaluation based on the U.S. Children’s Bureau “Child Maltreatment 2022” administrative statistics. The work looks at the following:
A. representation bias (distributions of victim demographics, maltreatment types, perpetrators, and caregiver risk factors in generated narratives vs. national statistics)
B. Semantic homogeneity (cosine similarities of SBERT embeddings of generated narratives vs. human-authored news)
C. Allocative bias (risk scores in controlled case varying only gender, race, or poverty)
Their findings show that LLMs over-represent female victims, under-represent infants, and over-emphasize physical/psychological abuse and medical neglect relative to neglect in national data. The work also shows how model outputs are highly homogenous compared to actual human news articles.

**Strengths:**

The primary strengths of the paper are as follows:

A. The authors look at a very underexplored direction in LLM evaluation. The need for a sociotechnical approach to evaluation has been discussed within the community, but this work showcases it with a relevant example.

B. The authors three-part methodology to examine representational analysis, semantic homogeneity, and allocative vignette testing provides a multi-faceted view on structural bias. This triangulation is stronger than the single-metric approaches typically found in bias evaluation work.

C.  The dataset design (10 prompts × 20 runs × 4 models) and the annotation protocol are clearly explained, and statistical tests are appropriate for the reported hypotheses

D.  By grounding model behavior against administrative population data, the authors make a strong argument about the dangers of “monoculture” across LLMs when applied to sensitive decision domains. This is important yet again in the scope of sociotechnical NLP.

**Weaknesses:**

Given the strengths in positionality and importance of the work, especially in the field of bias in NLP Benchmarking, the work does contain some weaknesses that need to be addressed. They are as follows:

A. The paper’s conclusions about population-level outcome distortions feel very overstated relative to the presented results in the paper. The study uses one administrative dataset (Child Maltreatment 2022). It does not model how narrative misalignments propagate into actual real-world decision systems. There is a disconnect between the narrative presented through the results and the actual impact the results can have on society; hence, claims about policy-level impact remain speculative.

B. The 10 prompts used appear to have a fixed narrative framing (e.g., “case report” style), which presumably introduces tone biases into generated content. The decoding parameters (temperature, top-p, seed control) are also not fully described. Without ablations across these generation settings, it is unclear whether observed biases arise from the models themselves or from prompt design.

C.  Reporting a Cohen’s κ of 1.00 across multiple categorical fields is questionably high and suggests that either the schema is too coarse or adjudication steps were simplified. This needs to be looked into again.

D. The semantic homogeneity metric relies solely on SBERT embeddings and a single human baseline corpus (NYT articles). This introduces potential biases, as SBERT carries its own structural trends, and news articles tend to be stylistically homogeneous. Multiple embedding (e.g., E5, GTR) and varying human text corpora would make the conclusions about “monoculture” more robust.

E.  The paper ends with a few insights. Still, it does not propose or test potential mitigation strategies or provide a coherent discussion of what needs to be done next. Including design recommendations or mitigation strategies would make the work more actionable for both researchers and practitioners.

**Questions:**

Addressing the weaknesses mentioned would provide better understanding to how this paper can be impactful within the ICLR community.

---

### Official Review · Reviewer_w1St · 2025-10-31

**Soundness:** 2
**Presentation:** 2
**Contribution:** 2
**Rating:** 2
**Confidence:** 4

**Summary:**

The paper evaluates LLMs for tasks related to child maltreatment. They conduct three evaluations.
- First, they generate "descriptions" of child maltreatment (similar to clinical vignettes), extract tabular characteristics from these descriptions, and compare the rates of various groups (race, gender, etc.) to the U.S. distribution of child maltreatment demographics.
- Second, motivated by work on algorithmic monoculture, they look at the similarity of different outputs generated by the same LLM.
- Third, they use LLMs to produce predicted risk scores varying only poverty status.

**Strengths:**

The topic (child maltreatment) is understudied by the ML community, and deserves more empirical study, insofar as this is a task that is actually being performed with LLMs (though this was unclear).

**Weaknesses:**

1. The specific evals are not well-contextualized. Why is the distribution of demographics in the generated vignettes important to study - is generating vignette descriptions of child maltreatment something that people are doing with LLMs in real-world scenarios? Similarly, for risk prediction, this is a task that's been extensively studied by the ML community using standard tabular ML - is this something that LLMs are being used for now?
2. Similarly, it isn't justified why matching the U.S. distribution is the normatively correct thing to do. If we were trying to design a benchmark, is that what we want? The authors find, for example, that the LLMs generate more descriptions of maltreatment involving girls. What if most policy so far focuses on boys - then maybe we want the LLM to talk more about girls, to bring greater awareness? Basically, the authors do not justify why matching the U.S. statistics is the correct behavior.
3. The implication of Figure 9 is not clear.
4. In Figure 8, I'm not sure if this says anything about algorithmic monoculture. We would expect that when given the same prompt, sampling two responses from the same LLM has much higher cosine similarity than two random news articles about child maltreatment.

**Questions:**

see above

---

### Official Review · Reviewer_V5ct · 2025-10-31

**Soundness:** 3
**Presentation:** 3
**Contribution:** 3
**Rating:** 6
**Confidence:** 4

**Summary:**

This paper evaluates large language models in a public-sector, high-stakes context: child maltreatment. Using the U.S. Children’s Bureau’s Child Maltreatment 2022 report as a ground-truth benchmark, the authors compare model outputs to population statistics across demographics, maltreatment types, risk factors, and perpetrator relationships. They find systematic and cross-model divergences. Models overrepresent female victims and severe abuse while underrepresenting neglect, despite neglect being dominant in the benchmark. They also show strong semantic homogeneity across prompts and models, and in controlled “allocative bias” tests models’ predicted risk varies with poverty level. The authors frame these effects as a domain-specific “monoculture” misalignment and call for evaluations that track population-level distortions, not just generic accuracy.

**Strengths:**

- Uses a real, comprehensive federal dataset as ground truth, making comparisons meaningful for policy use.

- Clear empirical evidence of representation gaps, for example overrepresentation of female victims and race distribution shifts relative to national data.

- Goes beyond benchmarks to population-level outcomes, highlighting underrepresentation of neglect and overemphasis on severe abuse types.

- Measures semantic homogeneity across prompts and models, supporting the monoculture claim with statistical tests.

- Introduces a simple, reproducible setup for “allocative bias” probes that isolate variables like gender, race, and poverty.

**Weaknesses:**

- Narrow model set and limited families may not fully capture frontier systems or deployment setups. Results could be model-version sensitive.

- The allocative bias experiments use few stylized cases, which may limit external validity relative to messy casework.

- The study surfaces misalignment but provides few mitigation strategies or ablations to test causes, for example instruction style, temperature, or retrieval grounding.

- Monoculture is argued with homogeneity metrics, but links from homogeneity to real decision errors are not fully quantified end-to-end.

- Results are summarized as average gaps. There is no exploration of a diversity–accuracy frontier, coverage of the solution set, or whether anonymization collapses the model ensemble toward a narrower region that may trade off recall of minority but valid cases. Please check https://arxiv.org/abs/2505.18139

**Questions:**

- Do results hold under stronger controls, for example larger model families, retrieval-augmented prompts with the federal report, or varied decoding? If homogeneity and misrepresentation shrink, the monoculture claim weakens.
- Can targeted mitigations, such as calibrated priors from the benchmark or identity-aware expertise routing, reduce distortions without harming accuracy?
- Instead of a single gold distribution, can you evaluate against stratified golds by state, urbanicity, or time to test whether some divergences are context-calibrated rather than errors?
- How sensitive are the representation gaps to prompt wording, role instructions, and temperature? A robustness sweep could show whether effects are structural or prompt-induced.

---

### Official Review · Reviewer_7ukb · 2025-11-01

**Soundness:** 2
**Presentation:** 1
**Contribution:** 2
**Rating:** 2
**Confidence:** 4

**Summary:**

This paper provides the first empirical evaluation of LLMs in the context of child maltreatment analysis, moving beyond traditional benchmarks and accuracy metrics to examine population-level outcomes. Grounded in the Child Maltreatment 2022 Report by the U.S. Children’s Bureau, the research systematically assesses LLM performance on tasks related to child abuse and neglect. Findings reveal that LLMs over-represent certain demographics (e.g., female victims) and more severe forms of abuse (e.g., physical and psychological), while underrepresenting the most common form—neglect. Moreover, LLM-generated narratives show high homogeneity across models, suggesting a cross-model monoculture effect in which diverse LLM architectures produce biased and uniform outputs. This indicates a systemic issue in current LLM design and training that may distort real-world, high-stakes decision-making outcomes.

**Strengths:**

1. This paper is the first empirical study to evaluate how LLMs handle the high-stakes topic of child maltreatment, comparing their outputs to real-world national statistics the Child Maltreatment 2022 Report.
2. Through analysis, this paper reached some interesting findings.

**Weaknesses:**

1. The technical contribution of this paper is primarily analytical. While it provides a thorough analysis of existing LLMs' performance on the task, it does not propose solutions to the identified limitations.
2. The paper's layout and formatting could be optimized. For instance, Figure 8 should be centered for better readability.

**Questions:**

In Section 4.1, the paper uses two independent annotators to construct a structured dataset based on the collected outputs. Why not use LLMs to perform this task?

---

### Meta-Review · Area_Chair_vLmq · 2025-12-28

**Summary:**

Reviewers consistently pointed out several concerns around the motivation and justification of the experimental/analytical choices in the paper. Initial scores were low, and the authors did not offer a rebuttal. I recommend that the authors revise and resubmit their work.

**Reviewer Concerns:**

No rebuttal.

**Reviewer Scores:**

No rebuttal.

---

### Decision · Program_Chairs · 2026-01-26

Reject